# DELTA-AI: LOCAL OBJECTIVES FOR AMORTIZED INFERENCE IN SPARSE GRAPHICAL MODELS

**Jean-Pierre Falet**[*], **Hae Beom Lee**[*], **Nikolay Malkin**[*], **Chen Sun,**
**Dragos Secrieru, Dinghuai Zhang, Guillaume Lajoie**[†], **Yoshua Bengio**[◇]
Mila – Québec AI Institute, Université de Montréal
Montreal, Quebec, Canada
{jean-pierre.falet, hae-beom.lee, nikolay.malkin}@mila.quebec

## ABSTRACT

We present a new algorithm for amortized inference in sparse probabilistic graphical models (PGMs), which we call $\Delta$-amortized inference ($\Delta$-AI). Our approach is based on the observation that when the sampling of variables in a PGM is seen as a sequence of actions taken by an agent, sparsity of the PGM enables local credit assignment in the agent's policy learning objective. This yields a local constraint that can be turned into a local loss in the style of generative flow networks (GFlowNets) that enables off-policy training but avoids the need to instantiate all the random variables for each parameter update, thus speeding up training considerably. The $\Delta$-AI objective matches the conditional distribution of a variable given its Markov blanket in a tractable learned sampler, which has the structure of a Bayesian network, with the same conditional distribution under the target PGM. As such, the trained sampler recovers marginals and conditional distributions of interest and enables inference of partial subsets of variables. We illustrate $\Delta$-AI's effectiveness for sampling from synthetic PGMs and training latent variable models with sparse factor structure.
Code: https://github.com/GFNOrg/Delta-AI.

## 1 INTRODUCTION

Probabilistic modeling in high-dimensional spaces is challenging due to the combinatorially large number of possible modes. When the modes are not known *a priori* and are well-separated from each other, the convergence of local exploration algorithms such as Markov chain Monte Carlo (MCMC) is prohibitively slow. In contrast, amortized inference methods, which train models to perform approximate sampling from the distribution of interest, are potentially scalable to high-dimensional spaces and come with guarantees on generation time, but may also suffer from mode collapse issues. Generative flow networks (GFlowNets; Bengio et al., 2021) are a family of methods that have recently shown success in sampling distributions over high-dimensional discrete spaces. GFlowNets are amortized sequential samplers that are compatible with local exploration (Zhang et al., 2022) and allow stable off-policy training (Malkin et al., 2023), helping to overcome the obstacle of mode collapse.

A major limitation of GFlowNets is that they scale poorly in the length of the generative trajectory, which equals number of variables if they are sampled one at a time as in Zhang et al. (2022). This is because the (not necessarily normalized) target density – the GFlowNet's reward function – takes as input the instantiated values of *all* the variables. As a result, the sampling cost scales linearly in the dimension. In addition, because the learning signal for all steps in a sampling trajectory comes from the terminal reward, GFlowNets suffer from inefficient credit assignment when trajectories are long, unless the reward function can be meaningfully decomposed into partial rewards for partially instantiated variables (Pan et al., 2023).

Motivated by the GFlowNet methodology and its limitations, we propose $\Delta$-amortized inference ($\Delta$-AI), an algorithm for energy-based probabilistic inference and training that scales well with respect to the dimension of variables. $\Delta$-AI training recovers the conditional probability distributions in a Bayesian network that defines the same distribution as a target graphical model. $\Delta$-AI makes use of known graphical model structure to allow for local credit assignment: updating the parameters of the

---

[*]Equal contribution. [†]CIFAR AI Chair. [◇]CIFAR Senior Fellow.

amortized sampler requires only instantiating a single variable and its Markov blanket, in contrast to all variables. It can also be seen as a novel training algorithm for GFlowNets of a particular structure, where the GFlowNet policy performs amortized inference.

We summarize the advantages of Δ-AI as follows. First, Δ-AI enables significantly faster training than regular GFlowNets and other amortized inference methods because *i)* each training step can involve only a small subset of variables, so the sampling cost is negligible, *ii)* the local training signal is much stronger than that in regular GFlowNet methods, as it uses the decomposition of the energy function into terms, each of which is directly matched with the corresponding conditionals and marginals. Second, the memory cost is very low because computing each gradient update only involves a small subset of variables (for a sparse graphical model). Lastly, with Δ-AI we can benefit from flexible probabilistic inference by amortizing many possible sampling orders into a single sampler (by learning Bayesian networks with multiple Markov-equivalent structures), allowing inference over partial subsets of variables.

The paper is structured as follows:

- In §2, we introduce the necessary background on graphical models and present the GFlowNet approach to amortized inference as motivation.
- In §3, we define and analyze the proposed Δ-AI algorithm.
- In §4, we validate our idea on various synthetic energy-based models, showing that Δ-AI provides faster wall-clock convergence time compared to algorithms from prior work, with Δ-AI's convergence time even comparable to or better than that of (training-free) MCMC sampling, noting that the trade-off gets better if more samples are needed, as with amortized inference in general.
- In §5, we validate Δ-AI on the task of image generation using a latent variable model. We impose the inductive bias of a graphical model structure on the joint over observed and latent variables and use Δ-AI as the posterior sampler in an amortized variational expectation-maximization (EM) procedure.

## 2 BACKGROUND

### 2.1 PROBABILISTIC GRAPHICAL MODELS

In this section, we introduce the relevant notation and review the background on probabilistic graphical models (PGMs) that is essential to understand our main methodology. Details and further background can be found in §B.1 and in Koller & Friedman (2009). Additional related work is in §A.

We consider a collection of random variables $X = (X_v)_{v \in V}$ indexed by a set $V$. For concreteness, we will assume the $X_v$ are binary, so $X$ takes values $x \in \mathcal{X} = \{0, 1\}^{|V|}$. We also assume that $X$ has full support, *i.e.*, its density $p : \mathcal{X} \to \mathbb{R}$ is strictly positive. However, the entirety of the following discussion applies to any discrete spaces, and much of it generalizes to continuous random variables.

**Undirected graphical models (Markov networks / factor graphs).** Let $G = (V, E)$ be an undirected graph, with each vertex $v \in V$ corresponding to a variable $X_v$. $X$ is a *Markov network* with respect to $G$ if it satisfies the local Markov property:

$$X_v \perp\!\!\!\perp X_{V \setminus (\{v\} \cup \mathcal{N}_G(v))} \mid X_{\mathcal{N}_G(v)} \quad \forall v \in V,$$

where $\mathcal{N}_G(v)$ denotes the set of neighbours of $v$ in $G$ and, for $S \subseteq V$, $X_S$ denotes the projection of $X$ onto the variables indexed by $S$, $X_S = (X_v)_{v \in S}$. Markov networks are a natural way to encode known conditional independences in the joint distribution of the variables $X_v$.

A common way to specify of Markov networks is via a decomposition of the joint density into local *factors*, each depending on a subset of the variables. If $\phi_1, \ldots, \phi_K$ is a collection of functions taking positive real values, where $\phi_k$ depends on $X_{S_k}$ for some $S_k \subseteq V$ (*i.e.*, $\phi_k : \{0, 1\}^{|S_k|} \to \mathbb{R}_{>0}$), then the normalized product of these factors defines a density:

$$p(x) = \frac{1}{Z} \prod_{k=1}^{K} \phi_k(x_{S_k}), \quad x_S := (x_v)_{v \in S}. \tag{1}$$

This is an energy model with energy $\mathcal{E}(x) = -\sum_k \log \phi_k(x_{S_k})$. Note that the normalization constant $Z$ equals the sum of the product of factors over all $x \in \mathcal{X}$ and can be intractable. Distributions of the form (1) are equivalent to Markov networks with respect to a certain graph (see §B.1 for review).

**Directed graphical models (Bayesian networks).** Directed graphical models are a different way of encoding conditional independences among random variables. If $D = (V, A)$ be a directed acyclic graph (DAG), where $A \subset V \times V$ is the set of directed edges, then $X$ is a *Bayesian network* with graph

structure $D$ if its density $p$ is the product of the conditionals of each variable given its parents:

$$p(x) = \prod_{v \in V} p(x_v \mid x_{\text{Pa}(v)}), \quad \text{Pa}(v) := \{u : (u, v) \in A\}, \tag{2}$$

with the conditionals understood to be unconditional marginals if the parent set is empty. Note that each factor in (2) is normalized, unlike in (1). This property makes the joint distribution easy to sample: one chooses any topological ordering $v_1, v_2, \dots$ of $V$ consistent with the DAG $D$ and samples the $x_{v_i}$ in order, conditioning each $x_{v_i}$ on the (previously sampled) parents of $v_i$. In a Bayesian network, every variable is conditionally independent of its non-descendants given its parents.

**From Bayesian networks to Markov networks and back.** Any Bayesian network with respect to a DAG $D$ is also a Markov network, as every conditional in (2) can be considered a factor in a model of the form (1). However, the graph structure of the corresponding Markov network (called the *moral graph*) may have more edges than the underlying undirected graph of $D$, unless $D$ has no *immoralities* (induced subgraphs of the form $\bullet \rightarrow \bullet \leftarrow \bullet$).

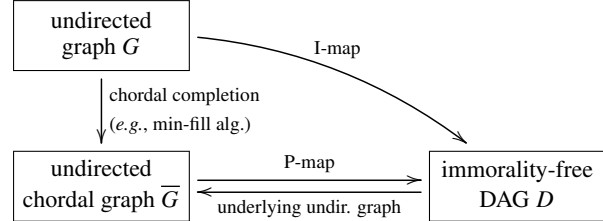

Conversely, if $X$ is a Markov network with respect to a graph $G$, then $X$ is a Bayesian network with respect to a directed graph with underlying undirected graph $G$ if $G$ is chordal (*i.e.*, has no cycle of length greater than 3 as an induced subgraph). Any graph $G$ can be made into a chordal graph $\overline{G}$ by adding edges (a process called *triangulation* or *chordalization*), which relaxes the conditional independence conditions on $X$. In turn, $\overline{G}$ has an immorality-free orientation $D$, and $X$ is a Bayesian network with respect to $D$. $D$ is called an I-map for $G$ and a P-map for $\overline{G}$.

Figure 1: Summary of the relationships between undirected graphs defining Markov networks (left) and DAGs defining Bayesian networks (right). Chordalization strictly relaxes the conditional independence constraints on a Markov network, while Markov networks with respect to a chordal graph and Bayesian networks with respect to its P-map are equivalent.

These relationships are summarized in Fig. 1 and more details are in §B.1.

## 2.2 Training and amortization in PGMs

**Maximum-likelihood training of PGMs.** The problem of generative modeling consists in optimizing the parameters of a PGM so as to maximize the likelihood of a given dataset of independent observations of the variables, $\mathcal{D} = \{x^{(i)}\}, x^{(i)} \in \mathcal{X}$.

For Bayesian networks, where the conditional distributions $p_\psi\left(x_v \mid x_{\text{Pa}(v)}\right)$ depend on a parameter vector $\psi$, the gradient of the log-likelihood is simply

$$\nabla_\psi \log p_\psi(\mathcal{D}) = \sum_i \nabla_\psi \log p_\psi\left(x^{(i)}\right) = \sum_i \sum_{v \in V} \nabla_\psi \log p_\psi\left(x_v^{(i)} \mid x_{\text{Pa}(v)}^{(i)}\right). \tag{3}$$

Generative modeling with energy-based models of the form (1), where the factors $\phi_k$ depend on a parameter vector $\psi$, is more difficult. The gradient of the log-likelihood of a sample is:

$$\nabla_\psi \log p_\psi(x) = \sum_k \nabla_\psi \log \phi_k\left(x_{S_k}\right) - \nabla_\psi \log Z, \quad \nabla_\psi \log Z = \mathbb{E}_{x \sim p_\psi(x)}[\nabla_\psi \log p_\psi(x)]. \tag{4}$$

Optimizing $\log p_\psi(\mathcal{D})$ thus consists of minimizing the energy energy for given samples $x^{(i)} \in \mathcal{D}$ (*positive phase* – first term in (4)) and maximizing it for samples from the model itself, $x \sim p_\psi(x)$) (*negative phase* – second term). The latter is complicated by the intractability of sampling the model exactly, motivating methods that draw approximate samples using MCMC (Hinton, 2002; Tieleman, 2008; Du et al., 2021) or an amortized inference method, such as a GFlowNet (Zhang et al., 2022).

One can also consider the case of generative modeling with *latent variables*, *i.e.*, where some subset $H \subset V$ of the variables is unobserved. The log-likelihood gradient for a partial observation $x_{V \setminus H}$ is

$$\nabla_\psi \log p_\psi\left(x_{V \setminus H}\right) = \mathbb{E}_{x_H \sim p_\psi(x_H \mid x_{V \setminus H})}[\nabla_\psi \log p_\psi(x)]. \tag{5}$$

Thus the objectives (3) and (4) can be applied to partial observations, as long as one can sample the conditional distribution over the latents $x_H$ given the observed values $x_{V \setminus H}$. This can be achieved using MCMC, importance sampling (Bornschein & Bengio, 2015; Le et al., 2019a), or amortized estimation of this conditional distribution, known as variational expectation-maximization (EM;

Dempster et al., 1977; Neal & Hinton, 1998; Koller & Friedman, 2009). A GFlowNet can also be used as the amortized estimator in variational EM (Hu et al., 2023).

**Amortized inference.** Complementary to the generative modeling problem in PGMs is that of amortized inference: given a factorized density $p$ of the form (1) with unknown normalization constant, one wishes to approximate $p$ by a distribution that can be tractably sampled. One option is to approximate $p$ by a sequential sampler, which iteratively chooses the value of one variable at a time conditioned on the values of those previously sampled. Training this sampler, as an agent that proposes action sequences and receives the unnormalized density as a reward, can be seen as a problem of policy optimization.

Given a density $p$ with factors $\phi_k$ and corresponding Markov network structure $G = (V, E)$, suppose that $D = (V, A)$ is an I-map for $G$, so $p$ is also a Bayesian network with respect to $D$. If $q_\theta$ is any Bayesian network with respect to $D$, with its conditional distributions given by parametric estimators $q_\theta(x_v \mid x_{\text{Pa}(v)})$, we are interested in fitting the conditionals of $q_\theta$ such that the distributions $q_\theta$ and $p$ are equal. If this is achieved, the sampler $q_\theta$ can be used to perform ancestral sampling of the variables in a fixed topological ordering $v_1, v_2, \ldots, v_{|V|}$.

### 2.3 Amortized inference with generative flow networks

We give some background on generative flow networks (GFlowNets), a group of deep reinforcement learning (RL) methods whose limitations in non-local credit assignment are a motivation for our proposed algorithms. Taking an RL view, the training of the conditionals in $q_\theta$ to sample from $p$ can be seen as a policy optimization problem. Ancestral sampling from $q_\theta$ is viewed as a sequence of actions taken by a sampling agent, which obtains a value of all the variables $x$ and receives a reward of $R(x) = \prod_k \phi_k(x_{S_k})$. The aim of training the policy is to make the likelihood of producing the sample $x$ proportional to the reward: $q_\theta(x) \propto R(x)$.

GFlowNets, which are a specialized family of reinforcement learning algorithms that train agents to match a target distribution, have yielded the state of the art in discrete probabilistic modeling problems (Zhang et al., 2022). However, in Zhang et al. (2022), the target density is unstructured; the sampler does not take advantage of a known factor structure and learns to sample the variables in an arbitrary order. Here we summarize, in simplified form, the GFlowNet objectives that can be used to train a Bayesian network as a sequential sampler that approximates the target distribution.

We describe TB and DB, two losses that achieve this. (We also use SubTB, a variance-reducing interpolation between TB and DB, in some experiments; see §B.2.)

**Trajectory balance.** To train $q_\theta$ as a GFlowNet with the trajectory balance (TB) objective (Malkin et al., 2022), one optimizes its parameters jointly with an estimate $Z_\theta$ of the normalization constant. The TB objective at a sample $x$ is

$$\mathcal{L}_{\text{TB}}(x) = (\log Z_\theta + \log q_\theta(x) - \log R(x))^2. \qquad (6)$$

If (6) is optimized to 0, so $q_\theta(x) = \frac{1}{Z_\theta} R(x)$, for all samples $x$, then $q_\theta$ is a perfect sampler for $p$ and $Z_\theta$ equals $Z$, the true normalization constant of $p$. The objective can be optimized by gradient descent using *on-policy* samples $x$ taken from $q_\theta$ itself, in which case it is equivalent in expected gradient to $D_{\text{KL}}(q_\theta \parallel p)$ (Malkin et al., 2023; Zimmermann et al., 2023), or at *off-policy* samples, such as those drawn from a tempered policy or from a known dataset, as done by Zhang et al. (2022); Zimmermann et al. (2023) in order to avoid mode collapse and thus better cover the target distribution.

**Detailed balance.** The detailed balance (DB) objective (Bengio et al., 2023) is an alternative loss that typically has lower variance but poorer performance than TB. The DB objective depends on an individual step in a sampling trajectory, rather than on the complete sample. Recall that $q_\theta$, as an agent, samples $x$ in a topological order: $x_{v_1}, x_{v_2}, \ldots, x_{v_{|V|}}$. To train $q_\theta$ with the DB objective, one also trains an auxiliary object, a *flow function* $F_\theta$ that outputs a scalar for any partially instantiated sample. The DB objective at the $i$-th step of the sampling trajectory is

$$\mathcal{L}_{\text{DB}}\left(x_{\{v_1,\ldots,v_{i-1}\}}, x_{v_i}\right) = \left(\log F_\theta\left(x_{\{v_1,\ldots,v_{i-1}\}}\right) + \log q_\theta\left(x_{v_i} \mid x_{\text{Pa}(v_i)}\right) - \log F_\theta\left(x_{\{v_1,\ldots,v_i\}}\right)\right)^2, \quad (7)$$

with the additional constraint that if $x$ is fully instantiated sample, then $F_\theta(x) = R(x)$. If the DB loss is optimized to 0 for every such transition, then $q_\theta$ is a perfect sampler for $p$, and the flow function at the initial state, $F(x_\emptyset)$, equals the true normalization constant $Z$. Just as with TB, one can flexibly choose the samples at which (7) is optimized by gradient descent.

Pan et al. (2023) found that DB performs strongly if the reward has a multiplicative decomposition that enables a form of reward shaping called the *forward-looking* parametrization of flows. In

our case of interest, the reward $R(x)$ has a multiplicative decomposition into the factors $\phi_k(x_{S_k})$. Rather than making $F_\theta$ a neural network taking a partial sample as input, one can learn $F_\theta$ as a multiplicative correction to a partially accumulated reward $\tilde{R}(x_{\{v_1,\ldots,v_i\}})$, *i.e.*,

$$F_\theta\left(x_{\{v_1,\ldots,v_i\}}\right) = \text{NN}_\theta\left(x_{\{v_1,\ldots,v_i\}}\right) \cdot \tilde{R}\left(x_{\{v_1,\ldots,v_i\}}\right), \tag{8}$$

where $\text{NN}_\theta$ is a neural network. We discuss two ways to define the partially reward $\tilde{R}$ in §B.2.

However, both the TB and DB losses are *non-local*: they require a fully instantiated sample $x$ to receive a training signal.

## 3 LOCAL CONSTRAINTS FOR MATCHING MARKOV AND BAYESIAN NETWORKS

In this section, we introduce $\Delta$-AI, which aims to address the limitations of GFlowNets in locality of credit assignment. We develop a novel GFlowNet-style objective, but whose loss only depends on a small subset of variables at a time. This is hypothesized to greatly improve training convergence and decrease resource requirements. We derive $\Delta$-AI by enforcing the equality of local conditional distributions in a pair of graphical models on the same set of variables.

**Setting.** We use the notation introduced in §2.1. Let $X = (X_v)_{v \in V}$ be a collection of discrete random variables whose density $p$ has a factor structure of the form (1), with factors $\phi_k$ depending on sets of variables $S_k$. Let $G = (V, E)$ be the graph with edges between any two variables that cooccur in a factor, so that $X$ is a Markov network with respect to $G$.

Let $\overline{G}$ be a chordal completion of $G$ and $D = (V, A)$ an immorality-free orientation of $\overline{G}$. By the results in §2.1, $X$ is a Bayesian network with respect to $D$, so $p$ has a factorization over the conditionals specified by $D$.

**$\Delta$-AI constraint.** The $\Delta$-AI constraint expresses the equality of the conditional distributions over one of the variables given its Markov blanket under the two factorizations, $p$ determined by the Markov network and $q$ by the Bayesian network. To be precise, suppose that $x, x' \in X$ are two settings of the variables that differ in exactly one variable $u$, *i.e.*, $x_u \neq x'_u$ and $x_{V \setminus \{u\}} = x'_{V \setminus \{u\}}$. Using the factorization for a Markov network (1) and that of a Bayesian network (2), and combining them appropriately, we have:

$$\prod_{k:u \in S_k} \frac{\phi_k\left(x_{S_k}\right)}{\phi_k\left(x'_{S_k}\right)} = \prod_{v \in \{u\} \cup \text{Ch}(u)} \frac{q\left(x_v \mid x_{\text{Pa}(v)}\right)}{q\left(x'_v \mid x'_{\text{Pa}(v)}\right)}, \qquad \text{Ch}(u) := \{v : (u, v) \in A\}. \tag{9}$$

We remark that the left side depends only on $x_{\{u\} \cup \mathcal{N}_G(u)}$, while the right side depends only on $x_{\{u\} \cup \mathcal{N}_{\overline{G}}(u)}$. Thus these constraints are *local* with respect to the PGM structure. They are nonetheless sufficient to recover the conditional distributions in the Bayesian network. The correctness and sufficiency of this constraint are formalized by the following proposition.

**Proposition 1.** *Suppose that $p : X \to \mathbb{R}_{>0}$ is the density of a Markov network with factors $\phi_k$ and that $q : X \to \mathbb{R}_{>0}$ is the density of a Bayesian network with respect to an I-map $D = (V, A)$. Then the following are equivalent: (1) for all $x, x'$ differing in a single variable $x_u$, (9) holds; (2) $p = q$.*

All proofs can be found in §D. We remark that the proof of Proposition 1, which uses the transitive closure of the single-variable mutation relation, is similar to the principle of *concrete score matching* (Meng et al., 2022), although the motivations are quite different.

**From constraints to losses.** We return to the problem of fitting a Bayesian network $q_\theta(\cdot \mid \cdot)$ to a given factorized model $p$, as in §2.2. Proposition 1 gives sufficient local constraints for $q_\theta$ to equal $p$. We can thus turn these constraints squared log-ratio losses in the style of (6) and (7):

$$\mathcal{L}_\Delta(x, u, x'_u) := \left[\sum_{k:u \in S_k} \log \frac{\phi_k\left(x_{S_k}\right)}{\phi_k\left(x'_{S_k}\right)} - \sum_{v \in \{u\} \cup \text{Ch}(u)} \log \frac{q_\theta\left(x_v \mid x_{\text{Pa}(v)}\right)}{q_\theta\left(x'_v \mid x'_{\text{Pa}(v)}\right)}\right]^2, \tag{10}$$

where $x'$ is defined as the perturbation of $x$ with $x_u$ changed to the value $x'_u$. Proposition 1 easily implies that if $\mathcal{L}_\Delta(x, u, x'_u) = 0$ for all $x, u, x'_u$, then $q_\theta$ is an amortized sampler for $p$.

**Training policy and exploration.** Just as in GFlowNet training (§2.3), the $\Delta$-AI objective $\mathcal{L}_\Delta(x, u, x'_u)$ can be optimized at on-policy samples $x \sim q_\theta(x)$ or those drawn from a tempered/dithered policy $q_{\tilde{\theta}}$ or dataset. The variable $u$ at which $x$ is perturbed is sampled uniformly in our experiments, although future work should investigate the effect of other perturbation policies. The full training algorithm is summarized in Algorithm 1.

**Local credit assignment.** We emphasize that (10) depends only on the values of $x$ in the $\overline{G}$-neighbourhood of $u$. The conditionals in this small subset of variables is given strong local supervision from only the factors that involve the variable $u$. This also significantly reduces the memory cost, as we do not have to maintain all variables in the computation graph to compute gradients.

**Masked autoencoder for amortizing all conditionals.** The form of the parametric model $q_\theta\left(x_v \mid x_{\mathrm{Pa}(v)}\right)$ is not specified in (10). In our experiments, we use a single neural network, a masked autoencoder (MAE), to model all the conditionals specified by $D$ simultaneously, allowing to make use of generalizable structure in the conditionals. To be precise, in the case of binary ($\pm 1$) data, $x_{\mathrm{Pa}(v)}$ is encoded as a $|V|$-length vector with all units except those corresponding to variables in $\mathrm{Pa}(v)$ set to 0, and the logit of $x_v$ is read off from the output unit corresponding to variable $v$.

**A stochastic loss.** If the number of terms on either side of (10) is prohibitively large, *e.g.*, due to $u$ having too many children added during chordal completion, an unbiased stochastic estimator of the gradient can be used. This estimator requires sampling only one child of $u$ at a time; see §E.

---

**Algorithm 1** $\Delta$-amortized inference (basic form)

**Require:** factors $\phi_k$, DAG $D$, model $q_\theta$, optimization/exploration hyperparameters
1: **repeat**
2:    Sample $x \sim \tilde{q_\theta}(x)$ (training policy)
3:    Sample $u \in V$, new value $x'_u \neq x_u$
4:    $\theta \leftarrow$ grad update w.r.t. $\mathcal{L}_\Delta(x, u, x'_u)$
5: **until** converged

---

**Amortizing over multiple DAG orders.** An advantage of the $\Delta$-AI formulation is that it allows to perform amortized inference in multiple I-maps (Bayesian network DAGs $D$) simultaneously. Indeed, the $\Delta$-AI constraint (9) is valid for any I-map $D$ for the Markov network $G$, and the same parametric model $q_\theta$ can be used for multiple DAGs (Fig. 2). Amortization of DAG orders is especially beneficial when using on-policy or tempered-policy training. If the loss is optimized only at variables near the root of the DAG, then *the downstream variables do not need to be sampled* to obtain a loss gradient (a direct consequence of the locality of the credit assignment), which reduces the overhead of computing 'rollouts' of the model to obtain samples for training. If the DAG order is freely chosen at each training iteration, then a given conditional will be trained as long as it is near the root of *some* I-map for the Markov network. While this procedure increases the number of functions that $q_\theta$ must learn to express, in practice, it does not substantially slow down convergence (see Fig. F.2), showing that the shared structure in the conditionals of $p$ may in fact aid learning.

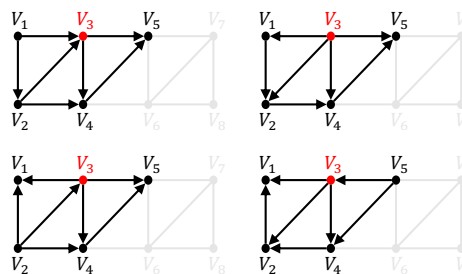

Figure 2: **Generating and amortizing multiple DAG orders.** The conditionals present two I-maps (DAGs) for the same undirected model $p$ are different: for example, the conditional $p(v_1 \mid v_2, v_3)$ appears in the two DAGs in the second row, but not in those in the first row. $\Delta$-AI learns a model $q$ that matches the conditionals in the target distribution $p$. If $q$ has a structure that allows taking varying subsets of variables as input, then it can be trained to match the conditionals appearing in multiple DAG structures simultaneously, and the resulting sampler can then be used for sampling in any of these DAGs.

**Uses of $\Delta$-AI.** $\Delta$-AI allows to solve both problems described in §2.2. If the parameters of the factors in a graphical model $p$, an amortized sampler $q_\theta$ can be trained using $\Delta$-AI to match the conditional distributions of $p$, thus amortizing the (generally intractable) sampling from $p$. If the parameters $\psi$ of the factors are unknown, then the parameters of $p_\psi$ can be updated using a maximum-likelihood gradient using samples from $q_\theta$, while $q$ can be trained to match the conditional distributions of a $p_\psi$ that evolves over the course of training, as we describe next.

**$\Delta$-AI in maximum-likelihood training of PGMs.** As discussed at the end of §2.1, amortized inference models can be used to obtain the log-likelihood gradient in generative modeling settings. We elaborate two such settings in which an amortized model trained using $\Delta$-AI can participate in a bilevel optimization loop with a generative model.

- In the training of an energy-based model, such as a factorized Markov network $p_\psi$, an amortized inference model $q_\theta$ can be used to draw the samples from $p_\psi$ needed in the negative phase in (4).
- In the training of a generative model $p_\psi$ on variables $V$ with latent variables $H \subset V$, an amortized inference model $q_\theta$ can be used to obtain the samples of $x_H$ conditioned on $x_{V \setminus H}$ needed in (5). If $p_\psi$ has a known Markov network structure $G$, then the conditional distribution of $x_H$ given $x_{V \setminus H}$ is a Markov network with respect to the induced subgraph of $G$ on $H$ and is therefore also

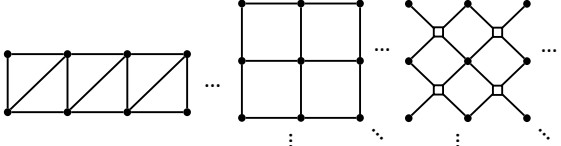

Figure 4: **Comparison of Δ-AI and GFlowNets**. Δ-AI converges to the target distribution fastest.

a Bayesian network with respect to an I-map for that subgraph. Thus one can train an amortized inference model $q_\theta$ as a sampler of latent variables $x_H$ using the Δ-AI objectives, where the model $q_\theta$ amortizes the dependence on the observed variables $x_{V \setminus H}$ by explicitly conditioning on them.

In both cases, one alternately optimizes $\theta$ with respect to the Δ-AI objectives and updates $\psi$ using (3) or (4). The locality of credit assignment also benefits this parameter-learning scenario, in which both the parameters of both $q_\theta$ and $p_\psi$ can be optimized after sampling only a single variable and its Markov blanket. The scheduling of the optimization steps in these bilevel optimization loops is an important design choice. Past work that used a GFlowNet as the amortized inference model performed alternating gradient steps with respect to $\theta$ and $\phi$ (Zhang et al., 2022) or used adaptive schedules based on the loss values (Hu et al., 2023). We describe our choices in the experiment sections below.

**Continuous Δ-AI and score matching.** A variant of Δ-AI with real-valued variables has connections to score matching; see §C for discussion and experiments.

## 4 EXPERIMENTS: SYNTHETIC DATA

In this section, we demonstrate the efficacy and efficiency of Δ-AI on sparse synthetic models in the case where the parameters of $p_\psi$ are known, and study the parameter-learning scenario in §5. We test the baselines and our method on the three graphical models shown in Fig. 3. See §F.1 for the detailed experimental setup.

**Δ-AI vs. GFlowNets.** We compare Δ-AI to regular GFlowNet losses: trajectory balance (TB) and detailed balance (DB). Both require all variables to be instantiated to compute a single reward, thus become inefficient when there is an excessive number of variables. We also compare against the forward-looking parameterization (FL-DB; see §2.3 and §B.2), which reparametrizes the flow function with intermediate energies to accelerate learning. All algorithms amortize over multiple DAG orders. The

(a) Ising ladder  (b) Ising lattice  (c) Factor lattice

Figure 3: **Graphical models.** (a) and (b) are UGMs for Ising models and (c) shows the factor graph model, where each factor is a small randomly initialized MLP with four arguments. (a) is chordal and (b,c) are non-chordal.

evaluation metric is negative log likelihood (NLL) of ground-truth samples from the target energy function (long-run Gibbs) with respect to each learned sampler.

Fig. 4 shows that our Δ-AI achieves significantly faster training convergence, because even with the same parametric model $q_\theta$ being learned, the baselines still need all the variables to be instantiated to receive a reward, resulting in longer wall-clock time and poor credit assignment. On the other hand, the Δ-AI objective requires only a small subset of variables to be instantiated, which results in much faster wall-clock time and stronger training signals for those small subsets. These results demonstrate the superiority of the proposed local objectives.

**Δ-AI vs. unstructured amortized inference.** In §F.1, we also compare Δ-AI with amortized samplers from past work that do not use the constraint of graphical model structure, finding that they tend to converge even slower than the structured variants (Fig. F.1).

**Δ-AI vs. MCMC.** We further compare Δ-AI against MCMC methods: Gibbs sampling and its variant Gibbs-With-Gradients (GWG; Grathwohl et al., 2021). The goal of this experiment is to measure the amortization costs and benefits by comparing wall-clock time to generate high-quality samples. In Fig. 5, the time axis of MCMC baselines means the wall-clock time of running 10k independent Gibbs chains, and for Δ-AI it is the training time. Note that for this experiment, we do not amortize multiple DAG orders in Δ-AI. Also note that the energy functions have a peaky

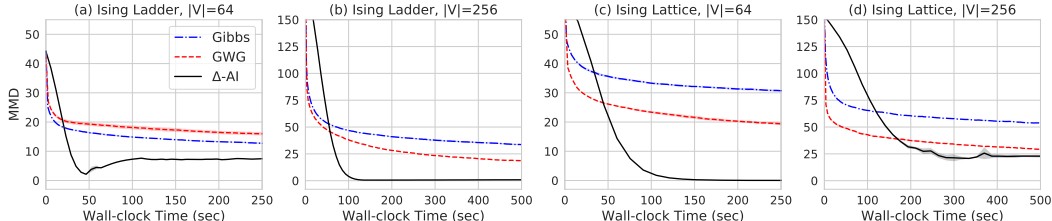

Figure 5: **Comparison against MCMC.** **(a, b)** chordal graph, **(c, d)** non-chordal graph. $\Delta$-AI provides a substantial amortization benefit, with training time smaller than the mixing time of MCMC chains.

landscape (see §F.1), so the MCMC baselines have difficulties in traversing all the modes. The evaluation metric is linear MMD between ground-truth samples (obtained by long-run Gibbs) and the samples generated from each amortized sampler. We can see from Fig. 5 that after a short period during which the amortized $\Delta$-AI sampler performs poorly (as the amortization network is close to initialization), soon $\Delta$-AI achieves significantly lower MMD, showing its superior effectiveness in traversing all the modes in the peaky distribution.

## 5 EXPERIMENTS: VARIATIONAL EM ON REAL DATA

To demonstrate how $\Delta$-AI can be used to learn a PGM from partial observations, we revisit the problem of latent variable modeling for MNIST images (Deng, 2012). Specifically, we want to train a generative model $p_\psi(x_H, x_{V \setminus H})$, where $x_H$ are latent variables and $x_{V \setminus H}$ are the observed pixel values. To train this model using variational EM, we learn an amortized variational posterior $q_\theta(x_H \mid x_{V \setminus H})$ using $\Delta$-AI or another inference method (E-step) and learn $p_\psi$ by maximizing $\log p_\psi(x_H, x_{V \setminus H})$ on samples from $q_\theta$ (M-step). Following past work on discrete probabilistic modeling, the images are discretized by interpreting the pixel values in $[0, 1]$ as parameters of independent Bernoulli random variables, from which we sample the pixel values.

We assume a pyramid-shaped graph structure for the latent variable model, shown in Fig. 6, in which dependencies between variables are local and sparse, and the observed variables are conditionally independent given the latents (see §F.2 for details). We define $p_\psi$ as a Bayesian network with a DAG structure that is an I-map for this pyramid-shaped undirected graphical model. The amortized posterior $q_\theta$, also a Bayesian network, has the same structure as $p_\psi$ over the subgraph consisting of latent variables, but has all edges from latent to hidden variables oriented upward, *i.e.*, conditioning the generation of latents in $H$ on observed variables in $V \setminus H$. This choice of DAG structure for $q_\theta$ ensures it is an I-map for $p_\psi$ conditioned on $V \setminus H$. We amortize $q_\theta$ over multiple DAG orders with the same underlying undirected graph, as described in §3.

Figure 6: **Structure of the PGM used in §5.** Grey layers above the pixel layer are composed of latent variables. Nodes within the boundaries of a blue square represent a clique with the node in the layer above (dashed lines), except in the pixel layer, where edges between pixels are removed to enforce conditional independence given the latents. Only three clique windows are shown for better visibility; in practice, windows are tiled across the layers with a specific window size and stride. The total number of binary latent variables in the three layers is 144 + 16 + 4 = 164. The graph contains only 8.9% of the edges possible in a model with conditionally independent pixels.

Given data $x_{V \setminus H}$, we train $q_\theta(x_H \mid x_{V \setminus H})$ to be proportional to $p_\psi(x_H, x_{V \setminus H})$, using the $\Delta$-AI objective (10) treating $p_\psi$ as a factorized model, or using a GFlowNet objective, using $p_\psi(x_H, x_{V \setminus H})$ as the reward for $x_H$. In both cases, we use a gently exploratory training policy: $x_H$ is sampled from a tempered $q_\theta$ with a low off-policy exploration rate (see §F.2).

We compare $\Delta$-AI with the following algorithms for approximating the posterior:

(1) **GFlowNets:** the objectives from §2.3 (**TB, DB, FL-DB**), trained with reward $p_\psi(x_H, x_{V \setminus H})$.
(2) **Mean-field variational EM:** a fully-factorized variational posterior trained using on-policy TB with the same reward, equivalent to minimizing KL between the amortized and true posteriors.
(3) **Wake-sleep:** maximizing $\log q_\theta(x_H \mid x_{V \setminus H})$ using one of the following: (i) **Sleep:** Unconditional samples $x$ from $p_\psi$ – the 'sleep' phase of wake-sleep (Hinton et al., 1995); (ii) **IW:** Imporance-weighted samples $x_H$ from the posterior given a dataset example $x_{V \setminus H}$, corresponding to the 'wake' phase of reweighted wake-sleep (Bornschein & Bengio, 2015). The samples are taken from $q_\theta$ and weighted by $\frac{p_\psi(x_H, x_{V \setminus H})}{q_\theta(x_H \mid x_{V \setminus H})}$ normalized over a batch.
(4) **Gibbs sampling:** sampling the posterior using $K$-step Gibbs sampling with respect to $p_\psi$.

Figure 7: **Results on latent-variable modeling of MNIST. (a)** Unconditional samples from $p_\psi$. For comparison, samples from a vanilla VAE (NLL ≈ 85, entropy ≈ 0.24), with an encoder and decoder parametrized using the same neural network architectures as for $q_\theta$ and $p_\psi$, respectively, and latent dimension equal to the number of latent variables in the graphical model, are shown at the top, and ground truth samples are shown at the bottom. **(b)** NLL of held-out test data for amortized methods. **(c)** Mean prediction entropy on unconditional samples of a pretrained MNIST classifier. All models are trained for 12 hours; mean ± std over 5 runs shown.

Algorithms (1), (2), and (3) use the same model architecture as Δ-AI for the conditionals in $q_\theta$, while (4) is not an amortized inference method and involves no learning in the E-step.

We use the NLL of test-set samples under $p_\psi$, estimated using the importance-weighted variational lower bound (Burda et al., 2016), as the main evaluation metric to assess convergence of the bilevel objective for the amortized inference algorithms. To assess the quality of generated samples, motivated by out-of-distribution detection methods (Liu et al., 2023), we also measure the mean prediction entropy of a standard pretrained MNIST classifier evaluated on unconditional samples from $p_\psi$.

Unconditional samples from $p_\psi$ and evaluation metrics are plotted in Fig. 7. As we move from a variational factorized posterior with learned structured prior (mean-field EM), to a structured posterior and structured prior (Δ-AI and GFlowNet baselines), the quality of unconditional samples improves. Wall-clock convergence for Δ-AI is quicker than all other baselines, with the IW variant closer to Δ-AI than the sleep variant, which is expected to be slower due to the initially poor quality of samples from $p_\psi$. While they performed well in this setting, wake-sleep variants are limited to settings with a normalized target density (unlike GFlowNets, MCMC, and Δ-AI) and scale poorly with the latent dimension. Gibbs sampling of the posterior, the standard non-amortized sampling algorithm in graphical models, is the third-slowest to converge. Although FL-DB uses the factorization of the energy function as an inductive bias for the state flow, it showed the slowest wall-clock convergence due to the expensive computation of the energy of every partially instantiated state.

Additional results are provided in §F.2; notably, we show that amortizing over multiple orders has barely any impact on convergence time for Δ-AI, as opposed to learning the conditionals of a single DAG (Fig. F.2), which not only demonstrates the benefit of parameter-sharing in $q_\theta$, but also Δ-AI's unique ability to do inference over partial subsets, which reduces sampling time. We emphasize that none of the baselines are capable of partial inference over subsets of variables. Δ-AI is therefore expected to lead to even more considerable improvements in wall-clock convergence time and memory requirements as the dimensionality of the graphical model is increased.

## 6 DISCUSSION

We have proposed an objective for amortized sampling in probabilistic graphical models that uses only local information as a learning signal. In this work, we evaluated our method in settings where the PGM structure was assumed known; however, in practice, the structure may need to be learned jointly with training of the amortized sampler by Δ-AI. We note that the related algorithm family of GFlowNets has successfully been applied to structure learning of *Bayesian* networks, both with and without joint inference of the parameters (Deleu et al., 2022; 2023; Nishikawa-Toomey et al., 2022), but not yet to structure learning of undirected graphical models.

A limitation of Δ-AI is the scaling to graphs for which chordalization results in very large Markov blankets, making training less efficient. Future work should thus consider methods such as stochastic losses (§E) or parametrization of small *joint* conditionals, such as those recovered by belief propagation in junction trees, rather than only the univariate conditionals specified by the Bayesian network. A related question is the choice of the subsets of variables to sample when learning from incomplete observations, which is necessary for learning on very large graphs where instantiating all variables is prohibitive. Heuristics for variable selection that maximize information gain – estimation of which can also be amortized – can be considered. Finally, Δ-AI should be applied to continuous spaces and on real-world data where we can impose the inductive bias of a PGM structure.

## REPRODUCIBILITY

Code is available at https://github.com/GFNOrg/Delta-AI.

## ACKNOWLEDGMENTS

The authors are grateful to Zhen Liu for discussions in the initial stages of this project, to Olexa Bilaniuk for help with code optimization, and to Thomas Jiralerspong for his help during the author response period.

JF acknowlegdes support from the FRQS/MSSS.

GL acknowledges funding from CIFAR, NSERC, Samsung, and a Canada Research Chair in Neural Computation and Interfacing.

YB acknowledges funding from CIFAR, NSERC, Intel, and Samsung.

The research was enabled in part by computational resources provided by the Digital Research Alliance of Canada (https://alliancecan.ca), Mila (https://mila.quebec), and NVIDIA.

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

## A    OTHER RELATED WORK

Amortized inference in graphical models has been proposed as a model for human reasoning (Gershman & Goodman, 2014). PGMs are the setting for many early variational inference methods (Saul et al., 1996; Beal, 2003; Jordan et al., 2004) and for related problems, *e.g.*, recovering tractable joints from unary conditionals such as those appearing in the Δ-AI objective (Arnold & Gokhale, 1998) and fitting conditionals over latent variables using approximate samples (Stuhlmüller et al., 2013; Simkus et al., 2023). Amortized inference is an ingredient in the training of structured generative models using variational EM (§2.1) and wake-sleep algorithms (Hinton et al., 1995; Bornschein & Bengio, 2015; Le et al., 2019b; Hewitt et al., 2020; Le et al., 2022). Beyond PGMs, amortized inference is used in weakly structured spaces, notably in variational autoencoders (Kingma & Welling, 2014; Burda et al., 2016; Rainforth et al., 2018); score-based models and diffusion models (Song & Ermon, 2019; Ho et al., 2020) can also be seen as performing amortized posterior inference.

## B    EXTENDED BACKGROUND

### B.1    GRAPHICAL MODELS

**Equivalence of factor models and Markov networks.**    A distribution of the form (1) is a Markov network with respect to a certain graph $G = (V, E)$. This graph is defined by the condition that $uv \in E$ if $u$ and $v$ cooccur in some factor's set of arguments $S_k$ (*i.e.*, $\exists k : \{u, v\} \subseteq S_k$). Remarkably, the converse is also true: the *Hammersley-Clifford theorem* (Hammersley & Clifford, 1971; Besag, 1974) states that if $X$ is a Markov network with respect to $G$, then its density has a factorization of the form (1), with each factor $\phi_k$ depending on a set of variables that forms a maximal clique (complete subgraph) in $G$.

**Conditional independences in Bayesian networks.**    The kinds of conditional independences that can expressed by a Bayesian network structure are distinct from those that can be expressed by a Markov network structure. For example, consider the DAG $D =$ $\boxed{1 \rightarrow 2 \leftarrow 3}$. Any Bayesian network with this structure has $X_1$ and $X_3$ marginally independent, but not necessarily conditionally independent given $X_2$. No Markov network structure can encode these constraints: $X_1$ and $X_3$ must be in different connected components (since they are marginally independent), implying their independence given $X_2$ as well.

We proceed to review the conversion from Bayesian to Markov networks and vice versa, elucidating the relationships that were summarized in Fig. 1.

**From Bayesian networks to Markov networks.**    If $X$ is a Bayesian network with respect to the graph $D = (V, A)$, then every conditional in (2) can be considered a factor in a model of the form (1). Thus $X$ is also a Markov network with respect to the graph $G = (V, E)$ that has an edge between two variables if they cooccur in some conditional (*i.e.*, $(u, v) \in E$ if $(u, v) \in A$, $(v, u) \in A$, or $(u, w) \in A$ and $(v, w) \in A$ for some $w \in V$). Equivalently, $G$ is constructed by forming a clique out of every variable together with its parents.

The underlying undirected graph of $D$ equals $G$ – the above procedure introduces no extra edges – if and only if $D$ has no *immoralities* (induced subgraphs of the form $\bullet \rightarrow \bullet \leftarrow \bullet$). Under these conditions, the set of distributions that are Bayesian networks with respect to $D$ coincides with the set of distributions that are Markov networks with respect to $G$.

**From Markov networks to Bayesian networks.**    Conversely, suppose that $X$ is a Markov network with respect to $G = (V, E)$. Recall that an *acyclic orientation* of $G$ is a DAG $D = (V, A)$ whose underlying undirected graph is $G$. It can be shown that $G$ has an acyclic orientation with no immoralities if and only $G$ is chordal.

Any graph $G$ can be converted into a chordal graph $\overline{G}$, known as a *chordal completion* of $G$, by adding extra edges. Although the optimal such chordalization is in general intractable to compute, the min-fill heuristic can be used to find chordal completions.

The set of distributions that are Markov networks with respect to $G$ is a subset of those that are Markov networks with respect to $\overline{G}$. The chordal completion $\overline{G}$ has an acyclic, immorality-free orientation $D$, which is called a *P-map* for $\overline{G}$ and an *I-map* for $G$. Any Markov network with respect to $\overline{G}$ is also a Bayesian network with respect to its P-map $D$. Consequently, any Markov network with respect to the original graph $G$ is a Bayesian network with respect to its I-map $D$.

**Sampling a P-map for a chordal factor graph.** Let $G$ be a chordal graph (note that any graph can be chordalized by inserting additional edges, for example, using the min-fill algorithm). We summarize how to find a P-map, *i.e.*, an orientation of the edges of $G$ that has no immoralities.

First, the order-maximal cliques are found using a maximum cardinality search algorithm. Let $C$ be the resulting set of maximal cliques. For maximal cliques $i, j$, let $S_{ij} = |C_i \cap C_j|$ be the number of shared vertices, and let $J = (C, S)$ be the weighted graph with a node corresponding to each clique and an edge of weight $S_{ij}$ between every pair of cliques $i$ and $j$. A *junction tree*, or clique tree, can be found by first sampling a maximum spanning tree for $J$ and then directing its edges away from any chosen root clique.

Finally, the orientation of $J$ is used to derive an orientation of $G$, by traversing the tree $J$ in a topological order, visiting the unvisited nodes within each clique in an arbitrary order, and orienting edges of $G$ towards any newly visited nodes.

The arbitrary choices made in this procedure – in the choice of maximum spanning tree, the choice of root clique, and the choice of ordering the unvisited nodes within each clique – can be varied to produce multiple P-maps for the same chordal graph.

### B.2 GENERATIVE FLOW NETWORKS

**Subtrajectory balance.** We describe an interpolation between the TB loss (6) and DB loss (7), known as SubTB (Madan et al., 2023), as it applies to our setting.

SubTB requires learning the same estimators as the DB loss: the flow function $F_\theta$, in addition to the conditionals. For a sequence of sampled values $x_{v_1}, x_{v_2}, \ldots, x_{v_{|V|}}$, the objective is decomposed over sub-ranges:

$$\mathcal{L}_{\text{SubTB},i:j}(x) = \left( \log F_\theta \left( x_{\{v_1,\ldots,v_i\}} \right) + \sum_{k=i+1}^{j} \log q_\theta \left( x_{v_k} \mid x_{\text{Pa}(v_k)} \right) - \log F_\theta \left( x_{\{v_1,\ldots,v_j\}} \right) \right)^2,$$

$$\mathcal{L}_{\text{SubTB}}(x) = \frac{\sum_{0 \le i < j \le |V|} \lambda^{j-i} \mathcal{L}_{\text{SubTB},i:j}(x)}{\sum_{0 \le i < j \le |V|} \lambda^{j-i}}.$$

Notice that $j = i + 1$ recovers DB and $i = 0, j = |V|$ recovers TB. The hyperparameter $\lambda$ controls the tradeoff between the losses for large $j - i$ (closer to TB) and small $j - i$ (closer to DB), although values of $\lambda$ close to 1 are typical.

The forward-looking parametrization (8) is also applicable to SubTB.

**Defining partial energies.** We experimented with two ways of defining the partial reward $\tilde{R}$ for partially instantiated variables. A natural way to define $\tilde{R}$ accumulates the product of the factors all of whose arguments are present in the partial sample:

$$\tilde{R} \left( x_{\{v_1,\ldots,v_i\}} \right) = \prod_{k:S_k \subseteq \{v_1,\ldots,v_i\}} \phi_k \left( x_{S_k} \right). \tag{11}$$

The partial reward signal is accumulated gradually as the arguments of more factors $\phi_k$ are 'filled in', providing a learning signal for partial samples.

However, we found that this option performs poorly in practice, possibly due to the parametrization of $\text{NN}_\theta$ in (8) as a head on the masked autoencoder (cf. §3) that also predicts the logits of conditionals in $q_\theta$. Such a model must learn to be sensitive to whether any given factor is fully instantiated (all entries nonzero – either +1 or −1), and we attribute the failure to this dependence being difficult to learn.

An alternative option, which works better in practice and is tested in our experiments, takes advantage of the factors – always MLPs or bilinear functions, in our experiments – being able to take arbitrary real-valued inputs, not just the discrete values in the sample space. Therefore, not-yet-sampled variables can simply be set to zero (the value corresponding to a masked input). We simply define

$$\tilde{R} \left( x_{\{v_1,\ldots,v_i\}} \right) = \prod_k \phi_k \left( x_{S_k}^{[v_1,\ldots,v_i]} \right), \tag{12}$$

where $x^{[v_1,\ldots,v_i]}$ denotes the masked sample that is the input the masked autoencoder, with 0 indicating absence of a value for all not-yet-sampled variables. Thus the factors for which not all values have been instantiated are computed with inputs of 0 for the missing variables.

## C  Δ-AI WITH CONTINUOUS VARIABLES

### C.1  CONTINUOUS Δ-AI AND SCORE MATCHING

Beyond the discrete spaces considered in this paper, it is interesting to consider the generalization of Δ-AI to continuous sample spaces, where the variables take real values. Recall that for two densities $p$ and $q$ on $\mathbb{R}^d$, the *Fisher divergence* is defined as

$$\mathbb{E}_{x \sim q(x)} \|\nabla_x \log q(x) - \nabla_x \log p(x)\|^2 = \mathbb{E}_{x \sim q(x)} \sum_i \left( \frac{\partial \log q(x)}{\partial x_i} - \frac{\partial \log p(x)}{\partial x_i} \right)^2. \quad (13)$$

The objective of score-based generative models (*e.g.*, Song & Ermon (2019)) optimizes (13) with respect to $p$ (or its score $\nabla \log p$) against a fixed distribution $q$ – typically the data distribution convolved with noise, which can be tractably sampled if a dataset is available. This is distinct from our motivation of matching $q$ to an intractable distribution $p$. However, we have the following result.

**Proposition 2** (informal). *In a PGM over real-valued variables, the expected Δ-AI objective (10) over x sampled from p, u chosen uniformly at random, and additive perturbations $x'_u = x_u + h$ approaches the score matching objective as $h \to 0$.*

Prop. 2 has implications for training generative models over continuous-valued data with known graphical model structure. It can motivate the use of scores in place of the perturbation ratios in (10) when working in continuous spaces: the scores share with the perturbation ratios the property of being independent of the variables outside the Markov blanket of $u$.

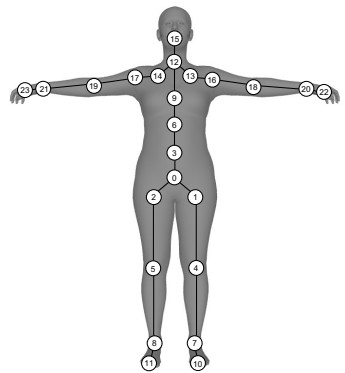

Figure C.1: **Illustration of the joint positions in the human poses dataset**. Each human pose is specified by 24 joints, as illustrated above, according to the SMPL body model (Loper et al., 2015). Each joint is specified by 3-dimensional joint angles of range $[-\pi, \pi]$, and connected to a set of other adjacent joints. The overall adjacency graph forms a tree, such that the graph is already chordal.

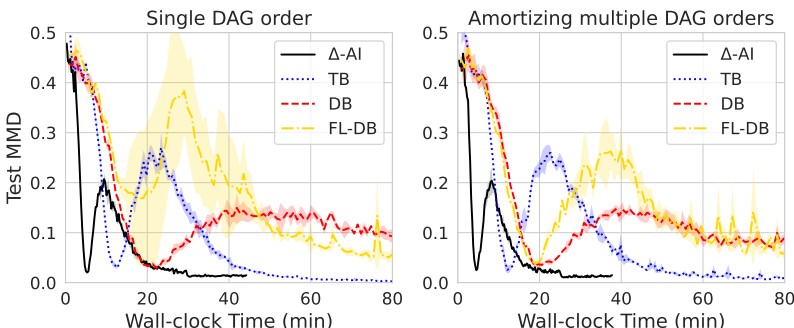

Figure C.2: **Experimental results** on the Poses dataset. We report mean and standard deviation over 3 runs.

### C.2  EXPERIMENT ON POSES DATASET

We further validate Δ-AI on a real-world dataset with continuous variables. This is a fully-observed setting with no latent variables.

**Dataset.**  We use the AMASS dataset (Mahmood et al., 2019), which is a large collection of recordings of real human poses. For this experiment, we focus on the KIT subset (Mandery et al., 2016),

and sample 100,000 instances from this subset. 20% is held-out as a test set. Each human pose is specified by 24 "joints", each of which represent a body part (neck, spine, left knee, right knee, etc.) that can take be articulated relative to adjacent body parts. See Figure C.1 for the illustration. Each joint orientation is specified by a 3-dimensional axis-angle representation in $[-\pi, \pi]^3$. The overall graph forms a tree, which is chordal.

**Factor graph and amortized sampler.** We assume that the data-generating factor graph has as adjacency matrix the structural connectivity of the joints in the dataset (i.e. if two joints are connected together by a body-part, such as the left wrist and left elbow, then these two variables will share an edge in the factor graph structure). Each factor takes 6 variables (2 joints × 3 angles) as input, and each factor is parameterized by a separate 3-layer multilayer perceptron with ReLU activations. The amortized sampler outputs the parameters of a Truncated Gaussian random variable (truncated to the range $[-1, 1]$ to model the conditional distributions in the factor graph. We found that optimizing truncated Gaussian distributions was more stable than Beta. We jointly train the sampler to approximate the factor graph distribution using Δ-AI and the factor parameters (the energy function) using (4).

**Baselines and evaluation metric.** We consider the same set of baselines as in Figure 4: TB, DB, and FL-DB. Note that for all the baselines, the sampling is done using the same sparse graphical model constraints (each variable is sampled conditioned only on their parents) for fair comparison. The evaluation metric is linear MMD between the test data and the generated samples from the amortized sampler.

**Results.** Figure C.2 shows that Δ-AI outperforms all the baselines by significant margin in terms of wall-clock convergence on the MMD. This tendency holds regardless of whether we learn to sample according to a single DAG I-map of the factor graph (Figure C.2, Left), or we amortizing multiple DAG orders using the same neural network (Figure C.2, Right).

## D  PROOFS

**Proposition 1.** *Suppose that $p : X \to \mathbb{R}_{>0}$ is the density of a Markov network with factors $\phi_k$ and that $q : X \to \mathbb{R}_{>0}$ is the density of a Bayesian network with respect to an I-map $D = (V, A)$. Then the following are equivalent: (1) for all $x, x'$ differing in a single variable $x_u$, (9) holds; (2) $p = q$.*

*Proof of Proposition 1.* **Part 1: (1) $\Longleftarrow$ (2).** Suppose that $q = p$, *i.e.*, the conditional distributions $q$ specified by the Bayesian network are the conditionals of the distribution $p$.

As stated before, the Δ-AI constraint stems from an equivalence of the Markov and Bayesian network factorizations.

Suppose that $x, x' \in X$ are two settings of the variables that differ in exactly one variable $u$, *i.e.*, $x_u \neq x'_u$ and $x_v = x_v$ for all $v \in V \setminus \{u\}$. Using the Markov network factorization (1), we have

$$\frac{p(x)}{p(x')} = \frac{\frac{1}{Z} \prod_k \phi_k \left(x_{S_k}\right)}{\frac{1}{Z} \prod_k \phi_k \left(x'_{S_k}\right)} = \prod_k \frac{\phi_k \left(x_{S_k}\right)}{\phi_k \left(x'_{S_k}\right)} = \prod_{k : u \in S_k} \frac{\phi_k \left(x_{S_k}\right)}{\phi_k \left(x'_{S_k}\right)}, \tag{14}$$

where the last equality uses that $x$ and $x'$ differ only at $u$, while the factor $\phi_k$ is independent of $x_u$ unless $u \in S_k$.

On the other hand, using the Bayesian network factorization (2), we have

$$\frac{p(x)}{p(x')} = \frac{\prod_{v \in V} p \left(x_v \mid x_{\text{Pa}(v)}\right)}{\prod_{v \in V} p \left(x'_v \mid x'_{\text{Pa}(v)}\right)} = \prod_{v \in V} \frac{p \left(x_v \mid x_{\text{Pa}(v)}\right)}{p \left(x'_v \mid x'_{\text{Pa}(v)}\right)} = \prod_{v \in \{u\} \cup \text{Ch}(u)} \frac{p \left(x_v \mid x_{\text{Pa}(v)}\right)}{p \left(x'_v \mid x'_{\text{Pa}(v)}\right)}. \tag{15}$$

The last equality similarly uses that the numerator and denominator can differ only if the conditional of $x_v$ given its parents depends on $x_u$, which occurs if and only if $v = u$ or $v$ is a child of $u$. The Δ-AI constraint (9) expresses precisely the equality of the two expressions (14) and (15).

**Part 2: (1) $\Longrightarrow$ (2).** Conversely, we must show that if the equality holds for all $x, x'$ differing in a single variable, then $p = q$. Because the left side equals of the equality equals $\frac{p(x)}{p(x')}$) by (14), and its right side equals $\frac{q(x)}{q(x')}$ by (15), the equality, together with positivity of all densities, implies that $\frac{p(x)}{q(x)} = \frac{p(x')}{q(x')}$.

The transitive closure of the relation $\sim$ on $X$ defined by $x \sim x'$ if $x$ and $x'$ differ at a single variable is the trivial equivalence relation, i.e., any $x, x' \in X$ are lined by a chain $x = x_1 \sim x_2 \sim \cdots \sim x_n = x'$ such that each $x_i, x_{i+1}$ differ in a single variable. It follows that for any $x, x' \in X$, $\frac{p(x)}{q(x)} = \frac{p(x')}{q(x')}$. Therefore, the two densities are proportional, which implies that they are equal. $\qquad\square$

**Proposition 2** (informal). *In a PGM over real-valued variables, the expected $\Delta$-AI objective (10) over $x$ sampled from $p$, $u$ chosen uniformly at random, and additive perturbations $x'_u = x_u + h$ approaches the score matching objective as $h \to 0$.*

*Proof of Proposition 2.* We assume the log-density $\log p$ is continuously differentiable. Let $e_i$ be the unit vector along coordinate $i$. The $\Delta$-AI objective is then

$$\mathbb{E}_{x\sim p(x)}\mathbb{E}_i \left( \log \frac{p(x+he_i)}{p(x)} - \log \frac{q_\theta(x+he_i)}{q_\theta(x)} \right)^2 = \mathbb{E}_{x\sim p(x)}\mathbb{E}_i \left( \frac{\partial \log p(x)}{\partial x_i} h - \frac{\partial \log q_\theta(x)}{\partial x_i} h + O(h^2) \right)^2$$

$$= h^2 \left( \mathbb{E}_{x\sim p(x)}\mathbb{E}_i \left( \frac{\partial \log p(x)}{\partial x_i} - \frac{\partial \log q_\theta(x)}{\partial x_i} \right)^2 + O(h) \right)$$

$$= \frac{h^2}{d} \left( \mathbb{E}_{x\sim p(x)} \|\nabla_x \log p(x) - \nabla_x \log q_\theta(x)\|^2 + O(h) \right).$$

We see that the $\Delta$-AI objective, normalized by $\frac{d}{h^2}$, approaches the gradient of the Fisher information (13) as $h \to 0$. $\qquad\square$

## E   STOCHASTIC LOSSES FOR SQUARED-SUM OBJECTIVES

**Proposition 3.** *Suppose $n > 1$. Let $S = \sum_{i=1}^{n} f_i$ and $L = \frac{1}{2}(g + S)^2$, where $g$ and each $f_i$ are functions of $\theta$. Let*

$$L_i := \frac{1}{2}(g + n\bar{f}_i)^2, \qquad L_{i,j} := \frac{n}{2}(\bar{g} + (n-1)\bar{f}_i + f_j)^2$$

*with $\bar{g}$ and $\bar{f}_i$ indicating that gradients are blocked, i.e., $\frac{\partial \bar{g}}{\partial \theta} = 0$ and $\frac{\partial \bar{f}_i}{\partial \theta} = 0$. Then*

$$\frac{\partial L}{\partial \theta} = \mathbb{E}_i \left[ \frac{\partial L_i}{\partial \theta} \right] + \mathbb{E}_{i\neq j} \left[ \frac{\partial L_{i,j}}{\partial \theta} \right], \qquad (16)$$

*where $\mathbb{E}_i[]$ is a uniform average over indices $i$ in $\{1, \ldots, n\}$ and $\mathbb{E}_{i\neq j}[]$ is a uniform average over all pairs of different indices in $\{1, \ldots, n\}$.*

*Proof.* The gradient of $L$ is:

$$\frac{\partial L}{\partial \theta} = (g + S) \left( \frac{\partial g}{\partial \theta} + \frac{\partial S}{\partial \theta} \right)$$

$$= g\frac{\partial g}{\partial \theta} + g\frac{\partial S}{\partial \theta} + S\frac{\partial g}{\partial \theta} + \sum_i f_i \frac{\partial f_i}{\partial \theta} + \sum_{i\neq j} f_i \frac{\partial f_j}{\partial \theta} \qquad (17)$$

Let us now show that we recover the same terms from the gradient of the right side of (16):

$$\mathbb{E}_i \left[ \frac{\partial L_i}{\partial \theta} \right] = \mathbb{E}_i \left[ (g + nf_i) \frac{\partial g}{\partial \theta} \right]$$

$$= g\frac{\partial g}{\partial \theta} + S\frac{\partial g}{\partial \theta},$$

$$\mathbb{E}_{i\neq j} \left[ \frac{\partial L_{i,j}}{\partial \theta} \right] = \mathbb{E}_{i\neq j} \left[ n(g + (n-1)f_i + f_j) \frac{\partial f_j}{\partial \theta} \right]$$

$$= g\frac{\partial S}{\partial \theta} + \sum_j f_j \frac{\partial f_j}{\partial \theta} + \sum_{i\neq j} f_i \frac{\partial f_j}{\partial \theta},$$

which exactly recovers the five terms in (17). $\qquad\square$

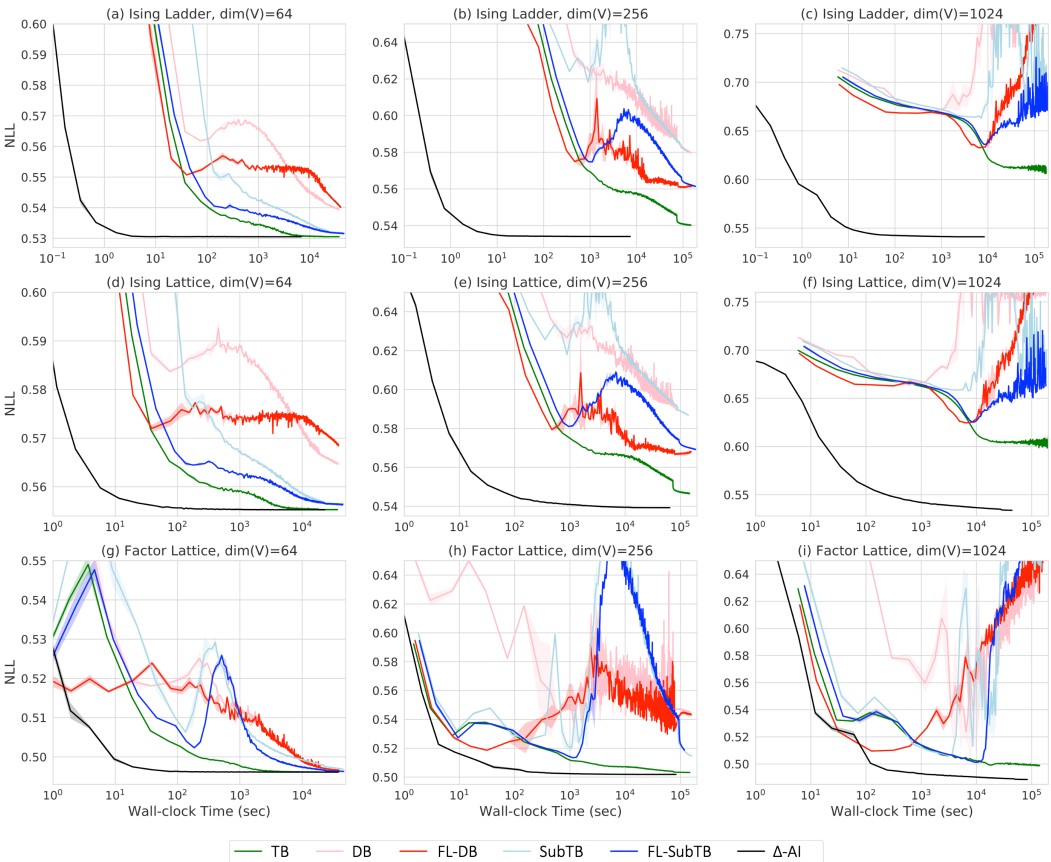

Figure F.1: **Wall-clock training convergence** on synthetic experiments with unstructured GFlowNet samplers from Zhang et al. (2022).

# F    EXTENDED EXPERIMENT DETAILS AND RESULTS

## F.1    SYNTHETIC EXPERIMENTS

**Energy models.**    In the synthetic experiments, we use either Ising models or factor graph models.

- The **Ising model** is a Markov random field, a distribution over $D$-dimensional binary vectors. Each element in a vector has a value of either $-1$ or $+1$. The following energy model $\mathcal{E}$ specifies the distribution over those binary vectors $x$:

$$p(x) \propto \exp(-\mathcal{E}(x)), \quad \mathcal{E}(x) = \sigma \cdot (-x^\top J x - x^\top b) \tag{18}$$

  where $J$ is a symmetric interaction matrix and $b$ is a unary potential vector. $\sigma > 0$ controls the sharpness of the energy function. Note that $J$ is sparse according to our assumption of sparse graphical model (see Fig. 3a and Fig. 3b). Each non-zero element of $J$ and $b$ has a value of either $-1$ or $+1$. In Figure 4 (a,b) we set $\sigma = 0.2$, and in Figure 5 we set $\sigma = 2$ (when $|V| = 64$) and $\sigma = 1$ (when $|V| = 256$).

- In the **factor graph model**, each factor is defined as a tiny MLP which has 1 hidden layer whose dimension is 10. Each MLP takes 4 arguments as an input and outputs a single scalar (see Fig. 3c). The parameters of those MLPs are randomly initialized with the spherical Gaussian distribution $\mathcal{N}(0, \sigma^2 I)$. We use $\sigma = 0.5$ in Figure 4 (c,d).

**Generating ground-truth Gibbs samples.**    We use NLL and MMD as an evaluation metric, which are based on the ground-truth samples from the target energy function. Therefore, it is crucial to have accurate ground-truth samples for correct evaluation. We generate those ground-truth samples by running 10k independent Gibbs chains for 10k steps. When the target energy functions are too peaky (in Fig. 5), we further anneal the energy function for the additional first 10k steps (i.e., initially we let the energy function be smooth and gradually lower the temperature during the first 10k steps) to prevent those chains from being attracted by only a few dominant modes.

**Model architecture and training details.** For all the GFN baselines and our method, the amortization network is an MAE of the following form: {Linear(512) → LayerNorm → ReLU} × 3 → Linear($|V|$). The other training details are as follows.

- **Δ-AI vs. GFNs (Fig. 4).** We consider $|V| \in \{64, 1024\}$. Each model is trained for total 200k iterations.

  - **Baseline GFNs:** Batchsize is set to 1k. We periodically sample a new DAG (every 50 iteration) to amortize over random DAG orders. Learning rate of the parameters of the amortized sampler is set to $10^{-3}$ and that of the partition function estimator is set to $10^{-1}$. Those learning rates are step-wisely decayed by 0.1 at 40k, 80k, 120k, 160k, and 180k-th iteration. For the training policy, we use $\epsilon$-uniform policy such that the sampling probability $p$ is defined as $p = (1 - \epsilon) \cdot p_{\text{on}} + \epsilon \cdot p_{\text{uniform}}$, where $p_{\text{on}}$ and $p_{\text{uniform}}$ are on- and uniform policy, respectively. We use $\epsilon = 0.1$.

  - **Δ-AI:** We periodically sample a set of sub-DAGs (every 50 iteration) to amortize over random DAG orders. For instance, when there are 64 variables (or nodes), we sample 16 sub-DAGs for each node (see Fig. 2), resulting in total $64 \times 16 = 1024$ batchsize. Learning rate of the parameters of the amortized sampler is set to $10^{-3}$, with the same learning rate scheduling as the baseline GFNs. For the training policy, we simply use the tempered off-policy with temperature set to 2.

- **Δ-AI vs. MCMCs (Fig. 5).** We consider $|V| \in \{64, 256\}$.

  - **Baseline MCMCs:** We run 10k parallel Gibbs chains in this experiment. At each evaluation step, we collect all those 10k samples and measure the MMD between the ground-truth samples.

  - **Δ-AI:** In this experiment, we do not amortize DAG orders because the flexibility of inference is not the focus of this experiment. Note that when the DAG order is fixed, we cannot use the efficient on-policy training as discussed in §3. However, we can simply only periodically sample the full dimensional on-policy samples to avoid the cost (e.g., every 10 iteration). Due to the peaky energy landscape, we found that gradually annealing the energy function for the first 10k steps helps stabilize the training. Also, for the first 10k steps, we use high temperature (e.g., 10) for better exploration and gradually lower the temperature to 1 (on-policy). We train total 10k steps, with the batchsize set to 10k and learning rate set to $10^{-3}$. Learning rate is step-wisely decayed at 2k, 4k, 6k, 8k, and 9k-th step.

**Δ-AI vs. any-order GFlowNets.** In past work, GFlowNets have been used as amortized samplers for unstructured energy-based models (Zhang et al., 2022). In that work, the GFlowNet sampler selects values for the variables in arbitrary order: while the models in §2.3 simply train the conditionals in the Bayesian network needed to sample the variables in a topological order, there, the generative policy selects both a variable and its value.

Fig. F.1 shows the training convergence on the synthetic experiments in the same way as Fig. 4. Among the baselines, TB shows the fastest and most stable convergence, whereas DB and SubTB show relatively unstable convergence, suggesting that the flow function is harder to train than forward transition probabilities. As a result, while FL variants (FL-DB and FL-SubTB) outperform DB and SubTB, they still underperform TB. Δ-AI shows much faster convergence.

All experiments are run with 5 random seeds, and the mean ± std regions are shown in the plots.

### F.2 LATENT VARIABLE MODELING OF MNIST

**Pyramid graph structure.** We expand on the specific pyramid-shaped graph structure used for the latent variable model in Fig. 6, which is used for all experiments. The first hidden layer has dimensions $6 \times 6$, the second $2 \times 2$, and the third is a single variable node. Each latent variable node is composed of 4 bernoulli random variables which are fully-connected. Each variable node in the first hidden layer form a clique with a window of $8 \times 8$ variable nodes in the pixel-layer below. This clique window reduces to $4 \times 4$, then $2 \times 2$, for the second and third hidden layers respectively. All edges in $V \setminus H$ are removed such that pixels are conditionally independent given $x_H$.

The resulting graph has 4608 edges between X and H (3.6% of all possible edges) and 7990 edges in the induced subgraph on $H$ (59.8% of all possible edges).

**Model architecture and training details.** The model architecture in all experiments that involves an amortized sampler $q_\theta$ (all except the Gibbs baseline) is a masked autoencoder (MAE), as described in §3. It consists of a shared trunk (3 fully-connected layers with ELU activations), and one output head (a linear layer) for each variable in $V$. Layer normalization and residual connections are

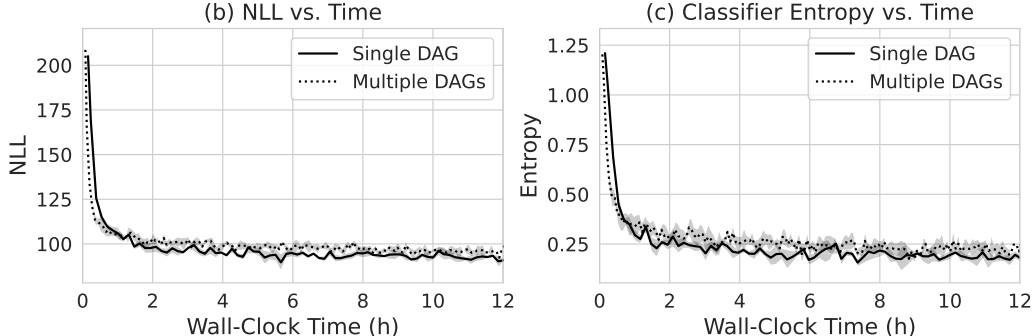

Figure F.2: **Comparison of Δ-AI trained for full inference on a single full DAG vs. partial inference on multiple DAGs.** All DAGs are I-maps for the underlying undirected pyramid-shaped graphical model (see Appendix F.2).

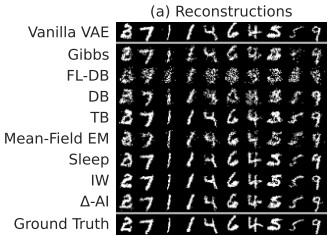

Figure F.3: **Reconstructions from Δ-AI and all baselines in § F.2**. The samples are compared to the ground-truth samples (bottom) that were used as input, and to a vanilla VAE (top).

used in the shared trunk. All layers are of dimension 512. We used this MAE to amortize conditionals, and used a separate vector of learnable parameters of size $|V|$ to infer the marginal for each variable.

We tuned the following hyperparameters:

- Learning rate for $q_\theta$ and $p_\psi$ in $[10^{-4}, 10^{-3}]$ for the conditionals, with the learning rate for the marginals fixed to the learning rate for the conditionals multiplied by a factor of 100.
- Temperature (only applicable to Δ-AI and the GFlowNet baselines, which use tempered samples for greater exploration) in $[1.5, 2, 3, 4]$. Tempered samples were drawn with probability $\epsilon \in [0.05, 0.1, 0.2, 0.5]$. In practice, we found Δ-AI to benefit from very light use of tempered sampling, with $\epsilon = 0.05$ and temperature 4. The GFlowNet baselines did best with larger values of epsilon.

Training of the bilevel objective was done in an alternating fashion: $q_\theta$ for 100 iterations, followed by $p_\psi$ for 100 iterations. For the Gibbs variant, the number of steps was set to 1; any larger number was prohibitively slow.

Mini-batch gradient descent was used along with the Adam optimizer. A step-wise learning rate decay (reducing the learning rate by half) was scheduled at 40%, 70%, and 90% of the total number of training iterations.

Batch size was 1148 for all experiments (which is the number of latents, 164, multiplied by 7). In the case of the Δ-AI variant that learns multiple orders by doing partial inference over subsets, an I-map for the sub-graph consisting of a variable and its Markov blanket was sampled for each variable (totalling 164 I-maps), and 7 samples were drawn from each I-map. For all other experiments pertaining to full inference, a single I-map for the full graphical model was sampled, and 1148 full samples were drawn. In the case of the Δ-AI variant trained for full inference on a single DAG (Fig. F.2), each of the 164 variables were perturbed 7 times in 7 different samples. For the experiments learning multiple DAG orders, a new DAG I-map (or set of DAG I-maps in the case of Δ-AI trained on partial instantiations) was sampled every 10 iterations. For the importance-weighted

variant, we used 7 independent importance-weighted samples per ground-truth sample (and reduced the batch-size by a factor of 7 to maintain the final batch-size equal to the other baselines).

We estimated NLL using the importance-weighted variational lower bound, with 100 independent importance-weighted samples per ground-truth sample. A standard pretrained MNIST classifier was used to compute the prediction entropy on unconditional samples from $p_\psi$. This classifier is a convolutional neural network with two convolution layers (10 channels, kernel size of 5, stride 1, followed by 20 channels, kernel size of 5, stride 1), each followed by a max-pooling layer and a ReLU activation, followed by two ReLU linear layers. Dropout (probability of 0.5) was applied after the convolutional block and between the two fully connected layers. Test accuracy was 0.9887.

