# OpenReview forum: "Delta-AI: Local objectives for amortized inference in sparse graphical models"
_ICLR.cc/2024/Conference — ICLR 2024 poster_

### Official Review · Reviewer_1ZeM · 2023-11-01

**Soundness:** 3 good
**Presentation:** 3 good
**Contribution:** 3 good
**Rating:** 8
**Confidence:** 2

**Summary:**

This paper presents Δ-amortized inference (Δ-AI), a new algorithm for efficient inference in sparse probabilistic graphical models (PGMs). Δ-AI leverages sparsity to assign credit locally in the agent's policy learning, enabling faster off-policy training without the need to instantiate all random variables. The trained sampler matches conditional distributions and recovers marginals, making it effective for inference in partial variable subsets. Experimental results demonstrate its efficiency for synthetic PGMs and latent variable models with sparse structures.

**Strengths:**

- The paper is well written and organized. It presents a clear and compelling motivation for the problem at hand. The discussion of prior works is thorough and well-structured, and the paper offers promising avenues for future research.
- The paper develops a novel algorithm namely, Δ-AI that offers substantially faster training when compared to regular GFlowNets and other amortized inference methods, since each training step involves only a small subset of variables, resulting in negligible sampling costs. Furthermore, the memory cost associated with Δ-AI is exceptionally low, since the algorithm leverages the sparsity of the probabilistic graphical model to compute the gradient update locally over a small set of variables. Lastly Δ-AI provides the advantage of flexible probabilistic inference by amortizing numerous potential sampling orders into a single sampler, by learning Bayesian networks with multiple Markov-equivalent structures.

**Weaknesses:**

As the current paper falls outside the scope of my research interests, I am unable to identify any significant weaknesses in the paper. Consequently, my confidence in assessing the paper is limited.

**Questions:**

- The paper hinges on the assumption that the factors of the distribution $\phi_{k}$ are known. This seems like a stringent condition, and it is worth exploring how the framework and algorithm proposed in the paper can be extended to scenarios where these factors are unknown and need to be learned.

-  The paper asserts that it accomplishes simultaneous amortization over multiple DAG structures. However, it would be beneficial to provide a more detailed explanation of how this simultaneous amortization is achieved.

---

> ### Author Response · Authors · 2023-11-17
> **Response to Reviewer 1ZeM**
>
> Thank you for your review. We appreciate your effort to understand the paper despite it being outside your scope of interests. We are happy that you found the motivation and structure of the paper clear, and we believe your summary and description of strengths are quite accurate.
>
> > The paper hinges on the assumption that the factors of the distribution are known.
>
> That is right, we assume the *factor structure* (the set of factors and their sets of arguments) are known. Their parameters are not necessarily known and can be learned with the aid of the amortized sampler (cf. the end of Section 2.1). We provide experiments in Section 5 where the task is to learn the parameters of the graphical model associated with a data-generation process, and demonstrate the benefit of using Δ-AI's partial inference capabilities to more efficiently learn factor parameters.
>
> Learning the undirected graphical model structure itself is quite a different problem and is outside the scope of this paper (and we are not aware of any amortized inference algorithm that jointly learns the factor structure, the parameters, and the amortized sampler). Indeed, it is typical for related work to study the problem of inference and learning in fixed graphical models, for example, using GFlowNets [Zhang et al., ICML 2022] or other algorithms taking advantage of sparsity (e.g., [Heap et al., "Massively parallel reweighted wake-sleep", UAI 2023]).
>
> We see two approaches possible to sampling the factor structure:
> - The first is to use a continuous relaxation of the space of "presence" variables for every possible factor (indicating whether a factor is present or not in the graphical model) and to fit the presence variables jointly with the factor parameters using an approximate method such as Gumbel-softmax. This would, however, not result in an unbiased sampler of the posterior over factor graph structures.
> - An approach more in line with the spirit of this paper would be to use a GFlowNet to sample the factor structure, perhaps by starting with an empty graph and inserting the factors one by one. This treats the factor structure as a latent variable and amortizes the sampling of the Bayesian posterior over this structure. We see this as a promising direction for future work, as GFlowNets have been successfully used to sample the structure of **directed** graphical models (Bayesian networks) ["Bayesian structure learning with generative flow networks", UAI 2022] and to sample continuous parameters associated to their factors ["Joint Bayesian inference of graphical structure and parameters", NeurIPS 2023].
>
> > The paper asserts that it accomplishes simultaneous amortization over multiple DAG structures. However, it would be beneficial to provide a more detailed explanation of how this simultaneous amortization is achieved.
>
> We attempted to explain this in Section 3, but welcome any concrete suggestions on how to make this clearer. Would it help to incorporate into the text a description along the following lines?
>
> "Figure 2 shows an example of multiple DAG structures (I-maps) for the same undirected graphical model. The conditionals present in the DAGs are different: for example, the conditional $p(v_1\mid v_2,v_3)$ appears in the two DAGs in the second row, but not in those in the first row. The proposed algorithm learns a model $q$ that matches the conditionals in the target distribution $p$. If $q$ has a structure that allows taking varying subsets of variables as input -- such as a masked autoencoder -- then it can be trained to match the conditionals appearing in multiple DAG structures simultaneously, and the resulting sampler can then be used for sampling in any of these DAGs."

---

> > ### Comment · Reviewer_1ZeM · 2023-11-23
> >
> > I appreciate the authors for addressing the concerns, and I am satisfied with the provided response. I retain the original score for the paper. Thank you.

---

### Official Review · Reviewer_iUke · 2023-11-01

**Soundness:** 3 good
**Presentation:** 2 fair
**Contribution:** 3 good
**Rating:** 6
**Confidence:** 2

**Summary:**

The authors propose a technique for amortized inference in sparse probabilistic models which they call \Delta-AI (\Delta-amortized inference) that takes advantage of sparse model structure (previously specified). This is done by matching the conditional distribution of a variable given its Markov blanket. The sparsity of the graphical model allows for local credit assignment in the policy learning objective. They experiment with synthetic PGMs and latent variable models with sparse factor graph structure to show the algorithms effectiveness.

**Strengths:**

The authors show how to take advantage of  known (or assumed) graphical model structure to allow for local credit assignment. Computationally, this lowers the memory requirement as parameter updates only requires instantiating a single variable and its Markov blanket.

**Weaknesses:**

The paper was hard to read - while there was a fair amount of discussion about the graphical model basics, there was not much about GFlowNets.

**Questions:**

Not having direct background / experience with GFlowNets, I found myself wondering why have two representations of the model (the
factorized Markov network p and the inference network q) - is the assumption that $p$ is provided (not just the structure / factors (sets of nodes) but also the exact parameters - or are the parameters of both learnt?

---

> ### Author Response · Authors · 2023-11-17
> **Response to Reviewer iUke**
>
> Thank you for your review. We appreciate the feedback and will use it to improve the paper. We answer your comments below.
>
> > The paper was hard to read - while there was a fair amount of discussion about the graphical model basics, there was not much about GFlowNets.
>
> The discussion in Section 2.2 provides a summary of the application of GFlowNets to the problem we consider (where a policy is equivalent to the collection of all the conditional probability distributions in the Bayesian network). This was done with the intention of making the paper self-contained and avoid a long digression in the main text. Moreover, we cite existing literature that serves as excellent introduction to GFlowNets, in an effort to provide the revelant tools to the interested but non-expert reader. Although we were certainly motivated by the limitations of GFlowNets as amortized inference algorithms in PGMs, the proposed algorithm does not rely on the general form of GFlowNets as models for amortized inference via sequential sampling, and Section 3 can be read independently of Section 2.2 (except its first and third paragraphs, which describe the shared mathematical setting).
>
> **Would the presentation be improved if 2.2 only introduced the problem setting (maximum likelihood training and amortized sampling in PGMs -- current end of 2.1 and beginning of 2.2) and the discussion of GFlowNets as a past approach were separated into a separate, self-contained Section 2.3?**
>
> We welcome any specific suggestions for how the background or any parts of the text could be improved.
>
> > Not having direct background / experience with GFlowNets, I found myself wondering why have two representations of the model (the factorized Markov network p and the inference network q) - is the assumption that is provided (not just the structure / factors (sets of nodes) but also the exact parameters - or are the parameters of both learnt?
>
> We assume the *factor structure* of $p$ (what the factors and their sets of arguments are) is known. We can solve different problems using our proposed algorithm, depending on whether the *parameters* of the factors in $p$ are known.
> - If the parameters of the factors are known, then $q$ can be trained using Δ-AI to match the conditional distributions of $p$. This question is interesting because sampling in undirected graphical models is generally intractable (see the experiments in Section 4).
> - If the parameters of the factors are unknown, then the parameters of $p$ can be updated using a maximum-likelihood gradient using samples from $q$ (see the end of Section 2.1 and experiments in Section 5), while $q$ can be trained using Δ-AI to match the conditional distributions of a $p$ that evolves over the course of training.
>
> We remark that these two settings do not rely on understanding of GFlowNets: both sampling in PGMs and training of factorized undirected graphical models are well-established problems outside the context of GFlowNets and the background is given in Section 2.1.
>
> Thank you again for your comments; please let us know if you have any further questions.

---

> > ### Comment · Reviewer_iUke · 2023-11-23
> >
> > Thanks for the clarifications.

---

### Official Review · Reviewer_mXkv · 2023-11-07

**Soundness:** 3 good
**Presentation:** 2 fair
**Contribution:** 2 fair
**Rating:** 6
**Confidence:** 3

**Summary:**

This work presents Delta-AI, an algorithm for the problem of amortized inference in PGMs. The main idea behind the paper is to leverage the sparsity of PGMs to pose a local constraint that can then be used as a novel loss function for GFlowNets. This local constraint makes GFlowNets more efficient both in terms of memory and time since only the few relevant variables can be used in each training step. The proposed inference algorithm is tested on a sytnetic data set as well as MNIST data and outperforms vanilla GFlowNets and other amortized inference algorithms.

**Strengths:**

1. Interesting use of local constraints in GFlowNets for amortized inference within the realm of probabilitic graphical models.

2. The proposed DELTA-AI seems sound and performs pretty well in the shown experimental settings.

**Weaknesses:**

1. The experiment settings are a bit weak in my opinion. Formalizing the idea and getting a a proof of concept is fine using a synthetic data. Also if there are no real data sets available that can also be considered but just considering MNIST as the real data is a bit limiting and does not show the full power of DELTA-AI. For example why not use the DELTA-AI loss within PixelVAE and try inference on natural images?

2. The paper is based on taking advantage of local credit assignment -> local losses this part gets kind of obsfuscated as the paper goes along. (but this is a simple fix by rewriting a bit of portions in sections 3)

**Questions:**

1. How will the proposed inference algorithm scale to natural images?


P.S: Being an emergency reviewer I might have missed some specifics and thus am lowering my confidence. Looking forward to the rebuttal.

---

> ### Author Response · Authors · 2023-11-17
> **Response to Reviewer mXkv**
>
> Thank you for your review. We are glad that you found the idea of Δ-AI interesting and value your constructive comments. We answer each of them below.
>
> > The experiment settings are a bit weak in my opinion.
>
> PixelVAE is an interesting suggestion, but large natural images are perhaps not the most appropriate setting, as it is difficult to impose a pre-determined sparse structure in the pixel space (we expand on this in our response to your question about scaling to natural images below). In fact, it is challenging to find in the literature a natural dataset with an undirected graph structure that is known (which is the setting we consider in this work), which explains why related publications studying inference methods typically experiment on synthetic tasks.
>
> We would like to emphasize that our contribution is primarily a novel method for amortized inference leveraging the assumption of sparsity. While we apply our trained amortized sampler to a task involving parameter learning in a PGM, our contribution is at the level of how the amortized sampler is trained rather than how the PGM parameters are learned.
>
> Nevertheless we acknowledge the reviewer's valuable comment and can look into applying Δ-AI to a natural dataset with known graph structure. We also refer the reviewer to our response to Reviewer 1ZeM, where we describe in more detail two possible future directions where the graph structure could be learned jointly with the amortized inference machine.
>
> > The paper is based on taking advantage of local credit assignment -> local losses this part gets kind of obsfuscated as the paper goes along. (but this is a simple fix by rewriting a bit of portions in sections 3)
>
> Thank you for your comment. We believe you mean that local credit assignment is not revisited after the paragraph "Local credit assignment" in Section 3. We will address this point later in Section 3 in the revised manuscript.
>
> > How will the proposed inference algorithm scale to natural images?
>
> Scaling to probabilistic graphical models (PGMs) in the context of natural image generation is a challenging task due to the sheer complexity of the high-dimensional graphical model. This is why the vast majority of existing literature constrain their study of inference methods in probabilistic graphical models to small dimensional settings. In fact, GFlowNets (which are used as a baseline in our parameter learning experiments) have, to our knowledge, never been used to learn an amortized sampler for a PGM applied to computer vision datasets larger than MNIST (see [Zhang et al., ICML 2022] and [Hu et al., ICML 2023] for two examples where discrete MNIST is the most complex task that is explored).
>
> Importantly, contrary to competing approaches, Δ-AI is expected to scale better to modeling natural images using a sparse PGM because of its partial inference/local credit assignment capabilities. Because our contribution is Δ-AI, an inference method, and we assume the graph structure is given, we chose to model MNIST images because MNIST images are expected to have strong local dependencies and a good fit can be obtained using a simple pyramid-shaped graph structure (see Figure 6). This graph structure may not provide the best fit for natural images, and a structure-learning algorithm would probably need to be used to obtain a graph structure before applying Δ-AI to train an amortized sampler on the corresponding graphical model. We will add discussion points in the revised manuscript about promissing future directions that could take advantage of the Δ-AI mechanism to address this.

---

> > ### Comment · Reviewer_mXkv · 2023-11-21
> > **Thanks for the clarifications**
> >
> > I would like to thank the authors for the comments. I do understand that the main contribution is a novel method for amortized inference but I still belive ethat the experiments do not do good justice to the claims. Anyhow I am happy with the response and lean towards acceptance. I thus keep my rating.

---

> ### Author Response · Authors · 2023-11-22
> **New experiment added on a natural dataset**
>
> We thank you once again for your comment. We have identified a dataset suitable to vailidate Δ-AI and have added it to Appendix B.2. The task is to learn to generate human poses from real data consisting of 24 joint variables. These are continuous random variables, so the experiment additionally demonstrates that Δ-AI can be used to learn continuous distributions. We used the structural connectivity of the joints (which joint is connected to which other joint by a body part) to define the adjacency matrix for the factor graph that we aim to learn the parameters of. Our experiment shows results that are consistent with our other experiments, in which Δ-AI (whether we amortize over multiple orders or learn a single DAG I-map for the factor graph) consistently converges faster than alternative methods.
>
> We hope this addition strengthens our experimental validation of Δ-AI.

---

### Author Response · Authors · 2023-11-22
**Revisions to the paper**

In response to the reviewers' comments, we would like to summarize the main changes we have brought to our paper, which are now included in the uploaded revision:

- Following Reviewer mxKv's comment that the impact of the locality of credit assignment was not clearly described later in Section 3, **we have added key sentences in the paragraph "Amortizing over multiple DAG orders" and "Δ-AI in maximum-likelihood training of PGMs"** to link back to this concept.

- **We have added discussion of structure learning to the Discussion section** in response to Reviewers mxKV and 1ZeM, to both make it clear that structure learning is a separate (and important) problem to solve that is not addressed in our paper (which is focused on a novel inference method), and also to suggest possible future directions that could use Δ-AI to jointly learn the graph structure.

- **We have changed the hierarchical structure of Section 2** by separating the background on GFlowNets (which is meant to serve as review of relevant prior work) from the background necessary to understand Δ-AI. We hope this helps clarify to Reviewer iUke that knowledge of GFlowNets is not necessary to understand Δ-AI, but rather puts Δ-AI in the broader context of other amortized samplers. We welcome any additional suggestions on how to further improve this section.

- **Section 3 now includes a more intuitive explanation of the amortization over DAG orders** (shown in Figure 2), as recommended by Reviewer 1ZeM.

- **We clarified in Section 3 the two main settings where we use Δ-AI**, which are 1) parameters of a graphical model (and its structure) are given, and the task is to learn a sampler that approximates the graphical model's distribution, and 2) only the structure is given, and the parameters of the graphical model are learned jointly with the sampler.

- Because the above changes lengthened the main text, we have moved the discussion of continuous Δ-AI and its connection with score matching to the appendix.

The changes are marked in blue in the updated pdf.

We thank the reviewers for their helpful suggestions and believe that the above modifications have strengthened our paper.

---

> ### Author Response · Authors · 2023-11-22
> **New experiment**
>
> Following Reviewer mxKV's suggestion to add a natural dataset to further strengthen our experimental validation, we have added a new experiment on generating human poses in Appendix B.2. We refer reviewers to our detailed explanation in our [latest response to mxKV](https://openreview.net/forum?id=LemSSn8htt&noteId=bf9WmjelZi) for further details.

---

### Meta-Review · Area_Chair_QVWE · 2023-12-11

**Metareview:**

This work presents a clever algorithm called delta-AI for inferences in sparse probabilistic graphical models. Ideas are experimentally checked in a reasonable way. The proposal seems sound and while experimental validation has limitations, it has been understood as sufficient for the purpose of this paper.

**Justification For Why Not Higher Score:**

Greater experimental analysis would make it stronger.

**Justification For Why Not Lower Score:**

Good ideas and reasonable results, research topic is interesting (and arguably growing).

---

### Decision · Program_Chairs · 2024-01-16

Accept (poster)